# Evaluating variable selection methods for multivariable regression models: A simulation study protocol

**Theresa Ullmann**[1], **Georg Heinze**[1], **Lorena Hafermann**[2], **Christine Schilhart-Wallisch**[1,3], **Daniela Dunkler**[1]*, **for TG2 of the STRATOS initiative**[¶]

**1** Institute of Clinical Biometrics, Center for Medical Data Science, Medical University of Vienna, Vienna, Austria, **2** Institute of Biometry and Clinical Epidemiology, Charité – Universitätsmedizin Berlin, corporate member of Freie Universität Berlin and Humboldt-Universität zu Berlin, Berlin, Germany, **3** Austrian Agency for Health and Food Safety (AGES), Vienna, Austria

¶ Membership list can be found in the Acknowledgments section.
* daniela.dunkler@meduniwien.ac.at

**Data Availability Statement:** This manuscript is a protocol of a simulation study. We intend to share the software code after the study has been

## Abstract

Researchers often perform data-driven variable selection when modeling the associations between an outcome and multiple independent variables in regression analysis. Variable selection may improve the interpretability, parsimony and/or predictive accuracy of a model. Yet variable selection can also have negative consequences, such as false exclusion of important variables or inclusion of noise variables, biased estimation of regression coefficients, underestimated standard errors and invalid confidence intervals, as well as model instability. While the potential advantages and disadvantages of variable selection have been discussed in the literature for decades, few large-scale simulation studies have neutrally compared data-driven variable selection methods with respect to their consequences for the resulting models. We present the protocol for a simulation study that will evaluate different variable selection methods: forward selection, stepwise forward selection, backward elimination, augmented backward elimination, univariable selection, univariable selection followed by backward elimination, and penalized likelihood approaches (Lasso, relaxed Lasso, adaptive Lasso). These methods will be compared with respect to false inclusion and/or exclusion of variables, consequences on bias and variance of the estimated regression coefficients, the validity of the confidence intervals for the coefficients, the accuracy of the estimated variable importance ranking, and the predictive performance of the selected models. We consider both linear and logistic regression in a low-dimensional setting (20 independent variables with 10 true predictors and 10 noise variables). The simulation will be based on real-world data from the National Health and Nutrition Examination Survey (NHANES). Publishing this study protocol ahead of performing the simulation increases transparency and allows integrating the perspective of other experts into the study design.

conducted and published. This will allow recreating our data and reproducing our simulation study.

**Funding:** This research was funded in part by the Austrian Science Fund (FWF, https://www.fwf.ac.at/en/) [I-4739-B] (for T.U. and C.W.) and by the German Research Foundation (DFG, https://www.dfg.de/en) [RA 2347/8-1] (for L. H.). For open access purposes, the author has applied a CC BY public copyright license to any author accepted manuscript version arising from this submission. The funders did not and will not have any role in study design, data collection and analysis, decision to publish, or preparation of the manuscript.

**Competing interests:** The authors have declared that no competing interests exist.

# 1 Introduction

Data-driven variable selection is frequently performed when modeling the associations between an outcome and multiple independent variables (sometimes also referred to as explanatory variables, covariates or predictors). Variable selection may help to generate parsimonious and interpretable models, and may also yield models with increased predictive accuracy. Despite these potential advantages, data-driven variable selection can also have unintended negative consequences that many researchers are not fully aware of. Variable selection induces additional uncertainty in the estimation process and may cause biased estimation of regression coefficients, model instability (i.e., models that are not robust with respect to small perturbations of the data set), and issues with post-selection inference such as underestimated standard errors and invalid confidence intervals [1–5].

A recent review [1] provided guidance about variable selection and gave an overview of possible consequences of variable selection. However, there are few systematic simulation studies that compare different variable selection methods with respect to their consequences for the resulting models (for some exceptions, see [6–10]). While many articles proposing new variable selection methods include a comparison with existing methods (based on simulated or real data), these comparisons are typically somewhat limited, often comparing the new method to only one to three competitors, even though there are many more existing methods. Moreover, these articles are inherently biased towards demonstrating superiority of the new methods. In particular, such studies cannot be considered as *neutral*. A neutral comparison study is a study whose authors do not have a vested interest in one of the competing methods, and are (as a group) approximately equally familiar with all considered methods [11, 12]. More neutral comparison studies about existing variable selection methods are needed to better understand their properties, a viewpoint that aligns with the goals of the STRATOS initiative (STRengthening Analytical Thinking for Observational Studies [13]). The STRATOS initiative is an international consortium of biostatistical experts, and aims to provide guidance in the design and analysis of observational studies for specialist and non-specialist audiences. This perspective motivates our comprehensive simulation study.

We will focus on *descriptive* modeling (i.e., describing the relationship between the outcome and the independent variables in a parsimonious manner) and *predictive* modeling (i.e., predicting the outcome as accurately as possible) [14]. Our setting is multivariable regression analysis with one outcome variable. The outcome is either continuous (linear regression) or binary (logistic regression). We simulate data in a low-dimensional scenario (20 variables consisting of 10 true predictors and 10 noise variables). Different variable selection methods with multiple parameter settings are compared: forward selection, stepwise forward selection, backward elimination, augmented backward elimination [15], univariable selection, univariable selection followed by backward elimination, the Lasso [16], the relaxed Lasso [9, 17], and the adaptive Lasso [18]. We compare the performances of these methods with respect to false inclusion and/or exclusion of variables, consequences on bias and variance of the estimated regression coefficients, the validity of the confidence intervals for the coefficients, the accuracy of the estimated variable importance ranking, and finally the predictive performance of the selected models.

Using simulated instead of real data allows us to a) know the true data generating process and b) systematically vary several data characteristics [19, 20]. For example, we will include varying sample sizes and $R^2$, as the consequences of variable selection depend on these parameters. To ensure that the simulation results are practically relevant, we use real data as the starting point for our simulation. The distributions and correlation structure of the variables are based on data from the National Health and Nutrition Examination Survey (NHANES) [21].

The choice of variables and true regression coefficients is inspired by an applied study about predicting the difference between ambulatory/home and clinic blood pressure readings [22]. Our simulated data thus mimics real cardiovascular data.

Our focus is on low-dimensional data, which is reflected in our simulation setting with twenty independent variables. Data of this type frequently appears in medicine and other application fields, and researchers often apply variable selection in this context. For example, a systematic review of models for COVID-19 prognosis [23, 24] identified 236 newly developed regression models for prediction. Data-driven variable selection was applied (and reported) for 196 models. In 165 models both the number of candidate predictors (i.e., the predictors considered at the start of data-driven selection) and the number of predictors in the final model were reported; the median numbers were 28 (range 4–130), and 6 (range 1–38), respectively. This demonstrates that low- to medium-dimensional data played an important role in COVID-19 prediction research. Of course, data-driven variable selection is also relevant for high-dimensional data. Comparing variable selection methods for high-dimensional data would require a different study design and is not the purpose of this planned simulation study.

As mentioned above, neutrality is an important goal when conducting systematic comparison studies. "Perfect" neutrality may be the ultimate goal, but this ideal can be difficult to achieve in practice. While we aim to be as neutral as possible, we disclose (for the purpose of full transparency) that one of the methods for variable selection included in our comparison, namely augmented backwards elimination, was originally proposed by two authors of the present study protocol [15]. Our goal was to not let this fact influence our choice of study design, though unconscious biases can never be fully excluded. Striving for as much neutrality as possible motivated us to publish this study protocol. This will allow us to integrate the comments of reviewers before performing the simulation. For the design of our study, results from previous smaller simulation studies and pilot studies were taken into account [1]; however, the study outlined in this protocol has not yet been run and analyzed. Preregistration of study protocols for simulation studies/methodological studies is still very rare (for an exception, see [25]). However, this practice could offer similar advantages to those discussed for preregistration in applied research, such as increased transparency and prevention of "hindsight bias" [26]. Potential advantages of preregistering protocols for simulation studies, but also possible limitations and challenges, are discussed more extensively elsewhere [27].

A specific goal of our simulation study is to evaluate previously published recommendations about variable selection [1], which we discuss in Section 2. We then describe our simulation design in Section 3, explain the planned code review in Section 4, and conclude the protocol with some final remarks in Section 5.

## 2 Previous variable selection recommendations

Varied viewpoints exist in the literature as to whether researchers should apply data-driven variable selection, and, if so, which methods and parameters are deemed preferable. Some authors generally caution against data-driven variable selection, stressing potential negative consequences [5]. Other authors put more focus on potential advantages of variable selection and are more optimistic about using selection methods, at least if the sample size is large enough and if selection is accompanied by a stability analysis [28]. In a review conducted by three co-authors of the present study protocol, Heinze et al. [1] summarized different perspectives from the literature. Drawing upon existing recommendations, but also taking their own experience and a small simulation study into account, they derived recommendations for the usage of variable selection methods. These recommendations consider both benefits and drawbacks of variable selection, thereby reconciling different viewpoints on the matter. The

recommendations depend on the "events-per-variable" (EPV) in the data. The EPV is the ratio between sample size (in linear regression) or the number of the less frequent outcome (in logistic regression) and the number of independent variables. Data-driven variable selection is applied on a carefully designed "global" model which includes all independent variables relevant for the research question. The denominator of EPV refers to the number of design variables (including possible dummy variables and other constructed variables) in this global model. The following bullet points list the recommendations, and how we plan to evaluate them.

- EPV > 25: While variable selection may generally work well for a large EPV value, the selection of independent variables with small effect size can still be unstable. If backward elimination is used, a stringent threshold of $\alpha$ = 0.05 or selection with the BIC may lead to a more accurate selection of variables than milder thresholds.
  *In our study*: We will check whether selection rates of variables with small standardized regression coefficients (e.g., ±0.25) are notably different from either 0 or 1 (which indicates instability). For backward elimination, we will evaluate whether the selection of variables is more accurate when using the threshold $\alpha$ = 0.05 or the BIC (which corresponds to even stricter thresholds for our considered sample sizes [1]), compared to using larger $\alpha$ values.

- 10 < EPV ≤ 25: In general, the selection of variables might be unstable with such an EPV. When variables with unclear effect size are selected, their effects might be over-estimated. Penalized estimation (Lasso) or postestimation shrinkage is thus recommended. If backward elimination is used, a threshold corresponding to selection with the AIC (approximately $\alpha$ = 0.157) is recommended, but not smaller $\alpha$ values.
  *In our study*: Again, we will evaluate stability by checking whether selection rates of variables, particularly those with small standardized regression coefficients, are notably different from either 0 or 1. We will also calculate the conditional bias (i.e., bias conditioned on selection) of the variables and analyze whether variables with small standardized regression coefficients have large conditional bias away from zero. For backward elimination, we will evaluate to which extent a threshold of $\alpha$ = 0.157 (or an even milder threshold of $\alpha$ = 0.5) selects the true predictors more frequently than smaller thresholds (i.e., a fixed threshold of $\alpha$ = 0.05 or selection with the BIC) [3].

- EPV ≤ 10: Data-driven variable selection is generally not recommended.
  *In our study*: We will analyze whether variable selection has negative consequences with respect to the different performance criteria.

The results of variable selection are not only influenced by EPV, but also by other aspects such as the $R^2$ of the model. We will thus consider different $R^2$ values in our simulation study. The recommendations above do not take $R^2$ into account, as the $R^2$ of the model is typically not known prior to the data analysis.

## 3 Simulation design

Morris et al. [19] proposed to describe the following components when reporting a simulation study: the aims of the study (A), the data-generating mechanisms (D), the estimands (i.e., the population quantities which are estimated) and other targets of interest (E), the methods to be compared (M), and the performance measures used for evaluating the methods (P). The ADEMP components of our study are briefly summarized in Tables 1 and 2. We now describe the components in more detail.

**Table 1. Summary of the simulation design, part 1: Aims and data-generating mechanisms.**

| Aims (Section 3.1) | Comparison of popular data-driven variable selection methods for multivariable linear or logistic regression, with respect to their consequences for the resulting models. |
| --- | --- |
| Data-generating mechanisms (Section 3.2) | • 20 variables: 10 predictors $X_1, \ldots, X_{10}$ and 10 noise variables $X_{11}, \ldots, X_{20}$ (mixture of binary and continuous variables) <br> • Distributions and correlation structure of the variables are based on NHANES data [21] (see Fig 1, S1 Fig and S1 Table). <br> • Standardized regression coefficients for the predictors $X_1, \ldots, X_{10}$: $(\beta_1^{sd}, \ldots, \beta_{10}^{sd}) = (1.5, -1, 1, 0.75, 0.5, 0.5, 0.5, -0.5, -0.25, -0.25)$ The regression coefficients for $X_{11}, \ldots, X_{20}$ are set to zero. <br> • For settings with linear effects, the outcome $Y$ is simulated as follows: <br> • For linear regression: $Y = x\beta + \epsilon$ with $\epsilon \sim N(0, \sigma^2)$, and $\sigma^2$ chosen such that $R^2 = 0.45$ (setting 1), $R^2 = 0.15$ (setting 2), or $R^2 = 0.7$ (setting 3). The intercept $\beta_0$ is set to 36. <br> • For logistic regression: outcomes $Y$ are drawn from a Bernoulli distribution with event probability $1/(1 + \exp(-c\, x\beta))$. The intercept $\beta_0$ and the constant $c > 0$ are adjusted such that <br> • the expected event probability equals 0.3 with Cox-Snell $R_{CS}^2 = 0.40$ (setting 4) or $R_{CS}^2 = 0.13$ (setting 5) <br> • the expected event probability equals 0.05 with Cox-Snell $R_{CS}^2 = 0.16$ (setting 6) or $R_{CS}^2 = 0.05$ (setting 7) <br> • To evaluate models with mildly misspecified functional forms (a frequent situation in practice), each of the 7 settings is also considered with $Y$ generated with nonlinear effects, yielding 14 settings in total (see Table 3). <br> • Additionally, we consider three simplified scenarios with all variables $\mathcal{N}(0, 1)$-distributed and/or uncorrelated (see Section 3.2.5 for details). <br> • Each setting is considered with varying sample sizes (see Table 4). |

## 3.1 Aims (A)

We aim to compare different variable selection methods for multivariable linear or logistic regression, with respect to their consequences for the resulting models. We consider consequences on bias and variance of the estimated regression coefficients, validity of confidence intervals for the coefficients, false inclusion or exclusion of variables, and predictive performance. We analyze the behavior of variable selection methods. . .

- . . . depending on sample size/EPV, with particular focus on evaluating the recommendations of Heinze et al. [1],

- . . . depending on the $R^2$ of the population model,

- . . . depending on the modeling goal (description or prediction),

- . . . when functional forms are misspecified (i.e., when fitting models assuming linear functional forms of continuous predictors even though the true functional forms are nonlinear),

- . . . when switching from our realistic scenario that mimics cardiovascular data to simplified scenarios (i.e., all variables are normally distributed and/or uncorrelated).

## 3.2 Data-generating mechanisms (D)

**3.2.1 Simulation of independent variables (predictors and noise variables).** We simulate 20 independent variables: 10 true predictors (from now on just called "predictors") and 10 noise variables. The correlation structure and distributions are based on real-world data from

**Table 2. Summary of the simulation design, part 2: Estimands, methods, and performance measures.**

| Estimands and other targets (Section 3.3) | Regression coefficients, model selection, predictions |
| --- | --- |
| Methods (Section 3.4) | • Forward selection with AIC<br>• Stepwise forward selection with AIC (forward selection with backward elimination steps)<br>• Backward elimination with $\alpha = 0.05$, $\alpha = 0.5$, AIC, or BIC<br>• Augmented backward elimination (ABE) with AIC and $\tau = 0.05$<br>• Univariable selection with $\alpha = 0.05$ or $\alpha = 0.20$<br>• Univariable selection with $\alpha = 0.20$ followed by backward elimination with $\alpha = 0.05$<br>• Lasso with λ tuned with 10-fold cross-validation<br>• Relaxed Lasso with λ tuned with 10-fold cross-validation or BIC<br>• Adaptive Lasso with λ tuned with 10-fold cross-validation<br>We also consider the global model with all variables. |
| Performance measures (Section 3.5) | Mainly for descriptive models:<br>• Bias and root of mean squared error of the coefficients<br>• Coverage and width of the 95% confidence intervals for the coefficients<br>• Type 1 error rate/power<br>• False positive rate/true positive rate<br>• Kendall's $\tau_B$ for variable rankings<br>• Selection rate of the true model, of an over-selection model, and of an under-selection model<br>Mainly for prediction models:<br>• Local bias and local root mean squared error w.r.t. estimated vs. true linear predictor<br>• Root mean squared error and median absolute error of estimated vs. true linear predictor, additionally AUC for logistic regression<br>• Integrated calibration index (ICI) |

the 2013–14 and 2015–2016 cycles of the National Health and Nutrition Examination Survey (NHANES) [21]. To choose suitable variables in the NHANES data, we drew inspiration from a regression model reported by Sheppard et al. [22] for predicting the difference between diastolic blood pressure readings as measured ambulatory/at home versus in the clinic. The variables are described in detail in S1 Appendix. The correlation matrix Σ for the simulation is based on the empirical correlation matrix of the variables. For better interpretability, we set correlations below 0.15 to zero and round all values to the closest multiple of 0.05 (see S1 Fig and S1 Table for the resulting correlation matrix).

To obtain distributions from the NHANES data, we fit Bernoulli distributions for the binary variables, and normal distributions, log-normal distributions, or approximations of the empirical cumulative distribution function (CDF) for the continuous variables. For each continuous variable, we truncate its distribution with the minimum of the variable in the NHANES data as the lower bound and the maximum as the upper bound. The resulting distributions are as follows (see also Fig 1):

• predictors: $X_1$ (log-normal), $X_2$ (continuous with approximated CDF), $X_3$ (log-normal), $X_4$ (binary, $p = 0.50$), $X_5$ (normal), $X_6$ (binary, $p = 0.29$), $X_7$ (log-normal), $X_8$ (log-normal), $X_9$ (normal), $X_{10}$ (binary, $p = 0.11$)

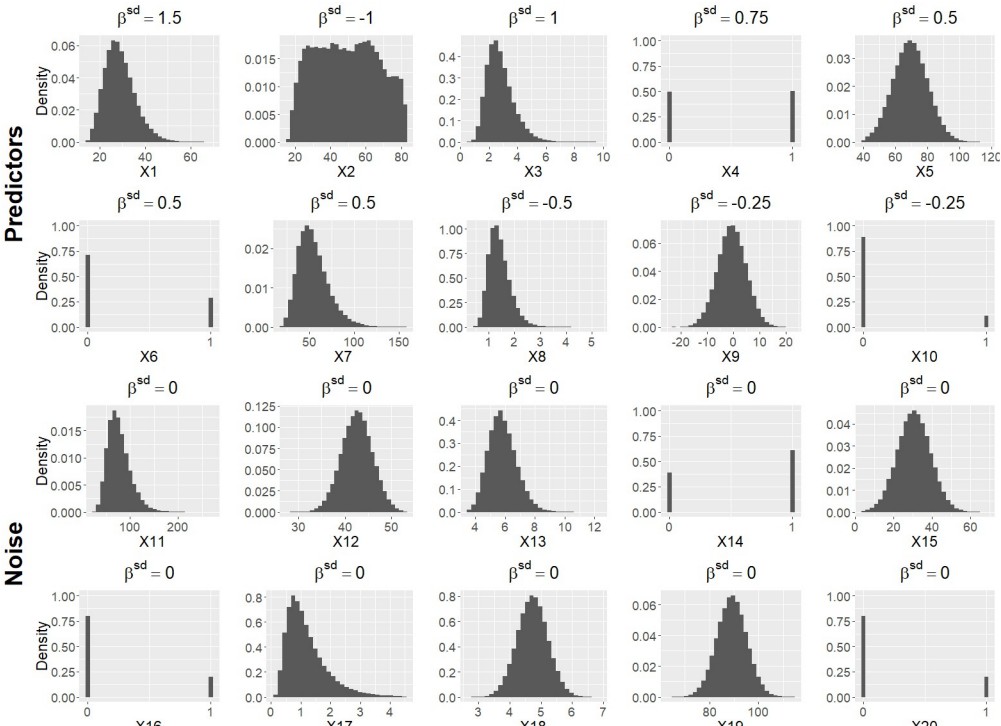

**Fig 1. Distributions and pre-specified standardized regression coefficients of predictors and noise variables.**
Predictors are ordered by absolute values of standardized regression coefficients. Histograms are based on a large
simulated dataset ($n = 100,000$).

- noise variables: $X_{11}$ (log-normal), $X_{12}$ (normal), $X_{13}$ (log-normal), $X_{14}$ (binary, $p = 0.61$), $X_{15}$ (normal), $X_{16}$ (binary, $p = 0.20$), $X_{17}$ (log-normal), $X_{18}$ (normal), $X_{19}$ (normal), $X_{20}$ (binary, $p = 0.20$)

The distributions, together with the correlation matrix $\Sigma$, are then used as input for the normal-to-anything (NORTA) method for simulation [29, 30].

**3.2.2 Choice of regression coefficients.** For choosing the standardized regression coefficients of the predictors $X_1, \ldots, X_{10}$, we drew inspiration from the coefficients reported for the regression model of Sheppard et al. [22]:

$$(\beta_1^{sd}, \ldots, \beta_{10}^{sd})^t = (1.5, -1, 1, 0.75, 0.5, 0.5, 0.5, -0.5, -0.25, -0.25)$$

This choice reflects a mixture of stronger and weaker effects, a situation typical for many applications in biology and medicine. We would expect different behaviors of the predictors during variable selection depending on their effects.

The standardized coefficients $\beta_j^{sd}$ are transformed into non-standardized coefficients $\beta_j$ as follows: standard deviations (SDs) of the variables are calculated based on a single large simulated dataset $D_P$ to approximate the population ($n = 100,000$) and the standardized coefficients are divided by these SDs.

The regression coefficients for the noise variables $X_{11}, \ldots, X_{20}$ are set to zero.

As intended, there is no systematic association between the absolute values $|\beta_j^{sd}|$ and the coefficients of determination $R_j^2$ for the regression of each variable $X_j$ on all other respective variables $X_l$, $l = 1, \ldots, 20$, $l \neq j$ (see S2 Fig). Moreover, five out of ten predictors have univariable effects that are larger than their multivariable effects. (Here, multivariable effects are obtained by fitting a model with all predictors and noise variables).

**3.2.3 Simulation of outcome $Y$.** The outcome $Y$ is simulated as follows:

- For linear regression: $Y = \boldsymbol{x\beta} + \epsilon$, with $\epsilon \sim N(0, \sigma^2)$, and $\sigma^2$ chosen such that $R^2 = 0.45$ (setting 1, main scenario), $R^2 = 0.15$ (setting 2, low $R^2$ scenario), or $R^2 = 0.7$ (setting 3, high $R^2$ scenario). The intercept $\beta_0$ is set to 36. The vector $\boldsymbol{x} = (1, x_1, \ldots, x_{10})$ denotes a simulated realization of the variables $X_1, \ldots, X_{10}$ with an added constant for the intercept.
  The required $\sigma^2$ values for obtaining $R^2 = 0.45$, $R^2 = 0.15$, or $R^2 = 0.7$ can be calculated as follows [9]:

$$\sigma^2 = Var(\boldsymbol{X_P\beta}) \frac{1 - R^2}{R^2},$$

  where $Var(\boldsymbol{X_P\beta})$ is calculated with the design matrix $\boldsymbol{X_P}$ obtained from the approximate "population dataset" $D_P$.

- For logistic regression: outcomes $Y$ are drawn from a Bernoulli distribution with event probability

$$\frac{1}{1 + \exp(-c\boldsymbol{x\beta})},$$

  with a constant $c > 0$.
  First, we set $c = 1$ and adjust the intercept $\beta_0$ manually such that the overall expected event probability equals either 0.3 or 0.05. The resulting Cox-Snell $R_{CS}^2$ values are 0.40 for event rate 0.3 and 0.16 for event rate 0.05. These values constitute the main settings for logistic regression, but note that they are different from the $R^2$ value of 0.45 in the main setting for linear regression (setting 1). Because setting 2 for linear regression considers 1/3 of the $R^2$ value in setting 1, we add analogous "low $R_{CS}^2$ settings" for logistic regression: $c$ and $\beta_0$ are adjusted to obtain 1/3 of the original $R_{CS}^2$ values (0.40/3 = 0.13 for event rate 0.3, and 0.16/3 = 0.05 for event rate 0.05). In contrast to linear regression (setting 3), we do not include an additional high $R^2$ setting: the maximum Cox-Snell $R^2$ values that are possible in theory are less than 1 (for event rate 0.3: approx. 0.71, for event rate 0.05: approx. 0.33), thus the $R^2$ values in the main settings can already be considered as relatively high.
  In summary, this yields the following settings 4–7, for which we also estimated the corresponding population areas under the receiver operating characteristic curve (AUC) based on the "population dataset" $D_P$:

- the overall expected event probability equals 0.3 with $R_{CS}^2 = 0.40$, AUC = 0.90 (setting 4) or $R_{CS}^2 = 0.13$, AUC = 0.73 (setting 5),

- the overall expected event probability equals 0.05 with $R_{CS}^2 = 0.16$, AUC = 0.94 (setting 6) or $R_{CS}^2 = 0.05$, AUC = 0.78 (setting 7).

**3.2.4 Nonlinear functional forms.** So far, we assumed that the functional forms of the effects of continuous predictors on $Y$ are linear. In applied studies in biology and medicine, the actual functional forms of such variables might often be nonlinear, but researchers

nonetheless fit a model with linear functional forms, e.g., because they are not aware that some functional forms might be nonlinear, or because they prefer a simpler model. To analyze the behavior of variable selection methods in this scenario, we include settings 1b-7b (corresponding to settings 1–7) where all predictors have *nonlinear* functional forms. The models that we consider for analysing the simulated data (linear/logistic regression) will not take the nonlinear functional forms into account and will thus be misspecified.

For each continuous predictor $X_j$, we define a function $g_j(x)$ that describes the nonlinear functional form of the effect of the predictor on $Y$. We choose various functional forms: quadratic, log-quadratic, exponential and sigmoid. The functions are depicted in S3 Fig; exact definitions are given in S1 Appendix.

The nonlinear composite predictor is then simulated as

$$\beta_0^{(g)} + \beta_1^{(g)} g_1(X_1) + \ldots + \beta_{10}^{(g)} g_{10}(X_{10})$$

Here, the coefficients $\beta_j^{(g)}$ (the letter $g$ alludes to the nonlinear transformations $g_j$) are chosen for continuous predictors such that

$$\beta_j^{(g)} SD(g_j(X_j)) \stackrel{!}{=} |\beta_j^{sd}| = |\beta_j| SD(X_j),$$

to obtain effects that are comparable in magnitude to the linear effects. The standard deviations $SD(X_j)$, $SD(g_j(X_j))$ are calculated based on the approximate "population dataset" $D_P$.

After determining $\boldsymbol{\beta}^{(g)}$, the outcome $Y$ is simulated as previously described in Section 3.2.3, with $\boldsymbol{x\beta}$ replaced by the nonlinear composite predictor.

The simulation settings 1–7 with linear effects and settings 1b-7b with nonlinear effects are summarized in Table 3. For settings 1b-7b, the $R^2$ values hold for the true model with nonlinear functional forms; the achieved $R^2$ values when functional forms are misspecified are expected to be lower. For logistic regression, using the coefficients $\beta_j^{(g)}$ for nonlinear effects in the main scenario yields slightly larger Cox-Snell $R_{CS}^2$ values compared to the analogous settings with linear effects: 0.43 for event rate 0.3 and 0.20 for event rate 0.05. For the low $R_{CS}^2$

**Table 3. Overview of simulation settings.**

| | linear regression | logistic regression | |
|---|---|---|---|
| | | event rate 0.3 | event rate 0.05 |
| main scenario... | | | |
| ...with linear effects | setting 1 | setting 4 | setting 6 |
| | $R^2 = 0.45$ | $R_{CS}^2 = 0.40$ | $R_{CS}^2 = 0.16$ |
| ...with nonlinear effects | setting 1b | setting 4b | setting 6b |
| | $R^2 = 0.45$ | $R_{CS}^2 = 0.43$ | $R_{CS}^2 = 0.20$ |
| low $R^2$ scenario... | | | |
| ...with linear effects | setting 2 | setting 5 | setting 7 |
| | $R^2 = 0.15$ | $R_{CS}^2 = 0.13$ | $R_{CS}^2 = 0.05$ |
| ...with nonlinear effects | setting 2b | setting 5b | setting 7b |
| | $R^2 = 0.15$ | $R_{CS}^2 = 0.14$ | $R_{CS}^2 = 0.07$ |
| high $R^2$ scenario... | | | |
| ...with linear effects | setting 3 | | |
| | $R^2 = 0.7$ | | |
| ...with nonlinear effects | setting 3b | | |
| | $R^2 = 0.7$ | | |

scenario, $c$ and $\beta_0^{(g)}$ are adjusted to obtain 1/3 of the $R_{CS}^2$ values (0.43/3 = 0.14 for event rate 0.3, and 0.20/3 = 0.07 for event rate 0.05), analogously to the procedure for settings with linear effects.

For the global model in the settings with nonlinear effects, we will not only calculate the usual standard errors of the regression coefficients, but also robust standard errors [31], to check whether robust SEs improve the coverage of the confidence intervals. If robust SEs improve the coverage for the global model, it would be interesting to analyze whether this is also the case for models obtained by variable selection; however, combining robust standard errors with variable selection requires some further work and would go beyond the scope of the proposed study. For now, we will restrict the investigation of robust SEs to the global model for linear regression.

**3.2.5 Simplified settings.** While our main focus is on simulating variables of various distribution types (e.g., Bernoulli, normal, and log-normal) and with correlation matrix $\Sigma$ based on the empirical correlation matrix from the NHANES data (S1 Table), we are also interested in the behavior of the variable selection methods for data with simpler distribution-correlation structures. We thus consider the three following simplified scenarios:

1. The variables are multivariate normal and independent: $X \sim \mathcal{N}_{20}(\mathbf{0}, I_{20})$, with $I_{20}$ denoting the $20 \times 20$ identity matrix.

2. The variables are multivariate normal and correlated: $X \sim \mathcal{N}_{20}(\mathbf{0}, \Sigma)$, with $\Sigma$ denoting the correlation matrix as described above.

3. The variables have the same individual distributions as described in Section 3.2.1 (Fig 1), but are not correlated.

For each of these three scenarios, we will consider the settings 1–2 and 4–7 (i.e., the main scenario and low $R^2$ scenario) with *linear* effects. This yields 3 * 6 = 18 simplified settings. For logistic regression, using the coefficients $\beta_j^{sd}$ from Section 3.2.2. will yield Cox-Snell $R_{CS}^2$ values that are slightly different from those given in Table 3.

Depending on the results for settings 1b-2b and 4b-7b with nonlinear effects, we might additionally consider nonlinear effects for the simplified scenario 3 (variables not multivariate normal and not correlated).

**3.2.6 Sample sizes.** For linear regression, we consider eight different sample sizes: 100, 200, 400, 500, 800, 1600, 3200, and 6400. These sample sizes result when doubling sample size six times from 100. Additionally, the sample size 500 is included because it corresponds to EPV = 25, and this EPV value was specifically mentioned in the recommendations of Heinze et al. [1].

For logistic regression, we first choose sample sizes corresponding to EPV = 25: $n$ = 1667 (event rate 0.3) respectively $n$ = 10, 000 (event rate 0.05).The other sample sizes for logistic regression are chosen differently depending on the event rate.For event rate 0.3, sample sizes are "aligned" to the samples sizes $\{n_1^{(lin)}, \ldots, n_7^{(lin)}\} = \{100, 200, 400, 800, 1600, 3200, 6400\}$ of linear regression as follows: sample sizes $n_k^{(log)}$ for logistic regression are chosen such that at sample size $n = n_k^{(log)}$, the regression coefficients in the logistic regression have approximately the same standard errors as the regression coefficients in the linear regression at $n = n_k^{(lin)}$. Our procedure for aligning the sample sizes is described in detail in S1 Appendix.

Because this procedure is unstable for small event rates, we do not use the alignment based on standard errors for event rate 0.05. Instead, we choose sample sizes corresponding to the EPV values in linear regression.

**Table 4. Sample sizes and EPV values for linear and logistic regression.**

| linear regression | $n$ | 100 | 200 | 400 | *500* | 800 | 1600 | 3200 | 6400 |
|---|---|---|---|---|---|---|---|---|---|
| | EPV | 5 | 10 | 20 | *25* | 40 | 80 | 160 | 320 |
| logistic regression, event rate 0.3 | $n$ | 183 | 365 | 730 | *1667* | 1461 | 2922 | 5844 | 11,687 |
| | EPV | 2.75 | 5.48 | 10.95 | *25* | 21.92 | 43.83 | 87.66 | 175.31 |
| logistic regression, event rate 0.05 | $n$ | 2000 | 4000 | 8000 | *10,000* | – | – | – | – |
| | EPV | 5 | 10 | 20 | *25* | – | – | – | – |

The resulting sample sizes are displayed in Table 4. The numbers below the sample sizes indicate the corresponding EPV values. For event rate 0.05, we will first include sample sizes only up to 10,000 (EPV = 25) to save computation time. We expect the variable selection methods to behave similarly for both event rates (0.3 and 0.05). If we observe different behaviors for event rate 0.05, we will include the additional sample sizes.

In S1 Appendix, we additionally report expected shrinkage factors for each setting, based on sample size and $R^2$ [32, 33].

## 3.3 Estimands and other targets (E)

As estimands, we consider the true *regression coefficients* of the data generating models. As further targets, we are interested in *model selection* (e.g., whether the true model is selected) and *predictive performance* of the selected models.

For the settings with linear functional forms, the regression coefficient estimands are the coefficients $\boldsymbol{\beta}$ (respectively $c\,\boldsymbol{\beta}$ for logistic regression) as described in Sections 3.2.2 and 3.2.3. For the settings with nonlinear effects, we cannot take the coefficients $\boldsymbol{\beta}^{(g)}$ as defined in 3.2.4 as estimands, because our linear/logistic regression models will not take nonlinear functional forms into account and will thus be misspecified.

Instead, we consider two alternative versions of estimands based on two different linear approximations of the nonlinear functions. Recall that we simulate the nonlinear composite predictor as

$$\beta_0^{(g)} + \beta_1^{(g)} g_1(X_1) + \ldots + \beta_{10}^{(g)} g_{10}(X_{10}).$$

Our first option for the estimands is $\boldsymbol{\beta}^{(proj)}$:

$$\beta_0^{(g)} + \beta_1^{(g)} g_1(X_1) + \ldots + \beta_{10}^{(g)} g_{10}(X_{10})$$
$$\approx \beta_0^{(proj)} + \beta_1^{(proj)} X_1 + \ldots + \beta_{10}^{(proj)} X_{10},$$

where $\boldsymbol{\beta}^{(proj)}$ are the coefficients obtained by projecting the true model with nonlinear functional forms onto one with linear functional forms. This projection is approximated by using the dataset $D_P$ as a "surrogate" for the population and fitting a linear/logistic regression model with linear functional forms to the nonlinear composite predictor (for linear regression) or to the outcome $Y$ that was simulated based on nonlinear functional forms (for logistic regression).

As the second option, we consider $\boldsymbol{\beta}^{(AS)}$:

$$\beta_0^{(g)} + \beta_1^{(g)} g_1(X_1) + \ldots + \beta_{10}^{(g)} g_{10}(X_{10})$$
$$\approx \beta_0^{(AS)} + \beta_1^{(AS)} X_1 + \ldots + \beta_{10}^{(AS)} X_{10},$$

where $\boldsymbol{\beta}^{(AS)}$ are the "average slope" coefficients. In contrast to the "projected" regression

coefficients, here each variable is considered individually. Each nonlinear effect $g_j(x)$ is approximated as $\alpha_j x$, where $\alpha_j$ is the average slope of $g_j$ weighted by the density of the $j$-th predictor $X_j$.

More precisely, let $f_j$ be the density function of $X_j$ as estimated from the NHANES data (i.e., the density function that is used as input for the simulation, see Section 3.2.1). We aim to approximate the "average slope" integral

$$\int_{min(X_j)}^{max(X_j)} g_j'(x)f_j(x)dx$$

For this purpose, we construct a partition $x_1^j = \min(X_j) \leq \ldots \leq x_{1001}^j = \max(X_j)$ of the range of $X_j$ with equal sub-interval lengths $d_j = x_{k+1}^j - x_k^j$, where $\min(X_j)$, $\max(X_j)$ are obtained from the NHANES data. Then the integral is approximated by

$$\alpha_j = \sum_{k=1}^{1000} d_j g_j'(x_k^j)f_j(x_k^j)$$

Finally, $\boldsymbol{\beta}^{(AS)}$ is obtained by setting $\beta_j^{(AS)} = \beta_j^{(g)}\alpha_j$.

## 3.4 Methods (M)

### 3.4.1 Overview of variable selection methods.

We include the following methods:

- Forward selection with AIC: starting from the model containing only the intercept, variables are iteratively added to the model based on their capability to decrease the AIC when included.

- Stepwise forward selection with AIC (i.e., forward selection with backward elimination steps): like simple forward selection, this method starts from the intercept model and adds variables based on the AIC. However, in each step, re-exclusion of already selected variables is allowed, based on the capability to decrease the AIC when removed.

- Backward elimination with $\alpha = 0.05$, with BIC, with AIC, and with $\alpha = 0.5$: starting from the global model, variables are iteratively removed, either based on their capability to decrease the BIC/AIC when removed, or based on the $p$-values of their coefficients. We do not consider a stepwise variant of backward elimination with forward selection steps, following the recommendations of Royston and Sauerbrei [28, p. 32] who argue that allowing re-inclusion of removed variables in backward elimination is rarely relevant, while allowing re-exclusion of included variables may cause a notable difference for forward selection.

- Augmented backward elimination (ABE) with AIC [15]: backward elimination is combined with the change-in-estimate criterion [34, 35]. A variable that would be removed in backward elimination based on AIC may stay in the model if its removal would induce a large change in the estimated regression coefficients of the other variables that are currently in the model. As threshold for the standardized change-in-estimate, we choose $\tau = 0.05$. We will use the R package abe [36].

- Univariable selection with $\alpha = 0.05$ and $\alpha = 0.20$: a variable is selected if its regression coefficient in a univariable model is significant at level $\alpha$. While many authors have advised against using univariable selection [5, 37, 38], the method is still often used in practice, which is why we include it in our simulation study.

- Univariable selection with $\alpha = 0.20$, followed by backward elimination with $\alpha = 0.05$: frequently, researchers use this combination instead of using only univariable selection or only backward elimination [39, 40] However, the warnings against univariable selection still apply to the combination method.

- Lasso [16]: a penalty on the coefficients is added to the OLS criterion (linear regression) or the negative log-likelihood (logistic regression), causing shrinkage of the coefficients toward zero and setting some of them to exactly zero.

- Relaxed Lasso [9, 17]: variables are selected with the Lasso, but the shrinkage of the coefficients of the selected variables is relaxed by refitting the model with the selected variables without penalty.

- Adaptive Lasso [18]: first, the global linear/logistic model is fit, then a Lasso with variable-specific weights for the penalty is estimated. The estimates from the first step serve to get the variable-specific weights for the second step: the weights are calculated such that a variable with larger regression coefficient in the first step is penalized less than a variable with smaller regression coefficient.

  For all variants of the Lasso, we will use the R package `glmnet` [41]. The complexity parameter $\lambda$ will be tuned with 10-fold cross-validation (CV). As performance criterion for the prediction on test sets during CV, we use the mean squared error for linear regression and deviance for logistic regression. For the relaxed Lasso, we additionally consider tuning $\lambda$ with the BIC.

We also consider the global model with all variables.

**3.4.2 Firth correction in logistic regression.** In the models for logistic regression, separation may occur (i.e., perfect separation of events and non-events by a linear combination of covariates), particularly for small to medium sample sizes and low event rates [42]. In this case, at least one parameter estimate is infinite. While separation can be detected by linear programming [43], we found that in practice, a simple and robust check can be performed by inspecting the model standard errors of the regression coefficients. If at least one standard error is extremely large, this indicates separation. A possible solution to the problem of separation is to apply the Firth correction to obtain finite parameter estimates [42, 44].

In the simulation settings for logistic regression, we check for each individual simulated dataset whether separation occurs. In the case of separation, we apply the Firth correction (with the FLIC intercept correction [45] to obtain unbiased predictions), otherwise we use the standard logistic regression. When Firth correction is applied, confidence intervals for the regression coefficients are calculated based on the profile penalized likelihood, otherwise based on the profile likelihood.

We describe our procedure to check for separation based on the model standard errors of the coefficients in S1 Appendix.

## 3.5 Performance measures (P)

We organize the performance measures into three categories, based on which estimands/targets they pertain to. Formulas for all performance measures are given in S1 Appendix.

Performance measures for the **regression coefficients** as estimands include *bias* and *RMSE* $\cdot \sqrt{n}$ (root of expected mean squared error multiplied by $\sqrt{n}$) of the estimated regression coefficients, and *coverage* and *width* $\cdot \sqrt{n}$ of the 95% confidence intervals of the coefficients. (The RMSE of coefficients and width of confidence intervals are multiplied by $\sqrt{n}$ for better comparability across sample sizes.) Moreover, we consider the *power* for predictors and

the *type 1 error* for noise variables (i.e., whether the confidence interval for the respective regression coefficient contains zero), as well as the selection rates of the variables (i.e., whether the regression coefficients are zero): the *true positive rate* for predictors and the *false positive rate* for noise variables. We also include Kendall's $\tau_B$ [46] to measure the agreement of the estimated ranking of variables (defined by ordering the variables based on absolute values of the estimated standardized regression coefficients) with the "true" ranking of the variables (defined by ordering the variables based on absolute values of the true standardized regression coefficients).

For bias and RMSE $\cdot \sqrt{n}$ of coefficients, coverage and width $\cdot \sqrt{n}$ of confidence intervals, and type 1 error/power for variables, the calculation can be performed unconditionally or conditionally on selection. In the unconditional approach, the coefficients and their confidence limits for non-selected variables are set to zero, while the conditional approach includes only simulation runs where the specific variable is selected.

Performance measures for **model selection** as target include the *selection rate of the true model* consisting exactly of the ten predictors, the *selection rate of an "over-selection" model* which we define as a model including all predictors as well as at least one noise variable (previously called an "inflated" model [15]), and the *selection rate of any "under-selection" model* defined as a model not containing all predictors but possibly including noise variables (previously called a "biased" model [15]).

Finally, we use multiple performance measures for **prediction**. Predictive performance is evaluated on a large test dataset ($n_{test} = 10,000$). One test dataset is simulated for each simulation setting. Prediction is assessed locally, i.e., at each value of the true linear predictor (*local bias* and *local RMSE* $\cdot \sqrt{n}$), as well as globally with the *global RMSE* $\cdot \sqrt{n}$ (i.e., the root of the expected mean squared error of the estimated vs. true linear predictor multiplied by $\sqrt{n}$) and the *global MAE* (i.e., the expected median absolute error of the estimated vs. true linear predictor). For logistic regression, global predictive performance is additionally evaluated with the *AUC*, i.e., area under the receiver operating characteristic curve. For both linear and logistic regression, the calibration of the predictions is measured with the *integrated calibration index* (ICI) [47]. The ICI is defined as the mean distance of the predicted outcomes/probabilities to the corresponding points on the calibration curve.

The performance measures for the regression coefficients and for model selection are primarily relevant for descriptive models, while performance measures for predictive performance are mainly relevant for prediction models. However, a descriptive model may also be suitable for prediction; therefore, performance measures for prediction could also be relevant for descriptive modeling. Vice versa, in prediction models, aspects such as interpretability, fairness etc. often play an important role; researchers might thus consider performance measures such as bias of coefficients also for prediction models.

## 3.6 Monte Carlo errors and number of simulation runs

The number of simulation repetitions $n_{sim}$ must be chosen large enough to estimate the performance measures with suitable accuracy, i.e., the Monte Carlo errors of the measures must be acceptable. We use the coverage of the confidence intervals as reference measure. The Monte Carlo standard error for the coverage can be calculated with the formula

$$\sqrt{\frac{\widehat{cover}(1 - \widehat{cover})}{n_{sim}}},$$

where $\widehat{cover}$ is the coverage estimated via simulation [19]. If $n_{sim} = 2000$ and $\widehat{cover} = 95\%$, the

Monte Carlo SE is about 0.5%. If $n_{sim}$ = 2000 and $\widehat{cover}$ = $50\%$ (the worst-case scenario), the Monte Carlo SE is about 1%, which is still acceptable. Therefore, we plan to use $n_{sim}$ = 2000 in our simulation for all settings (provided that this will be computationally feasible). We will then calculate the Monte Carlo SEs for all performance measures.

## 4 Code review

To ensure reproducibility, as well as readability, the code will be checked by another researcher (a "code reviewer") who works at the same institute as the first, second and last author of this protocol, but was not involved in planning the study. After writing the code, the first author (T.U.) will hand over the code to the code reviewer, together with instructions for running the code as well as some partial results (using less than the full $n_{sim}$ = 2000 repetitions). The code reviewer will then check the plausibility of the partial results and provide feedback on the simulation code, focusing on a) data generation, b) the implementation of the compared models, and c) the implementation of the performance measures applied to these models. Once T.U. and the code reviewer have agreed upon a final version of the code, T.U. will re-run the partial results, and the code reviewer will check the complete computational reproducibility by re-running the code on another machine. This check for reproducibility is done on the partial results as the generation of the final results is expected to require large amounts of computational resources. Once the reproducibility check has successfully concluded, T.U. will perform the full $n_{sim}$ = 2000 repetitions to generate the final results.

## 5 Final remarks

Our simulation study will enable researchers to better understand the consequences of variable selection, and will clarify differences in the performance of different selection methods depending on the considered scenarios. To make the results of the study more accessible and interpretable, we plan to display all results in an interactive web app (Shiny app) that will be published alongside the main paper. We will also make our code available on a Git repository, and will specify random seeds to ensure reproducibility of the results.

The performance measures for our study (Section 3.5) are defined as expected values and probabilities. Their estimation by simulation thus always involves taking the mean over (a part of) the simulation repetitions. However, if one only calculates the mean over the repetitions, one might miss relevant properties of the distribution of the values over the simulation repetitions. We will thus use distribution plots and correlation analyses to evaluate the simulation results in more detail [19]. Moreover, we will analyze how many variables were selected by each variable selection method. We did not include model size as a performance measure in Section 3.5 because there is no clear target value and smaller/larger values are not automatically better/worse (a smaller model size is preferable in some applications, but might be less relevant in others). A specific focus on model size (e.g., comparing different variable selection methods under constraints w.r.t. the number of chosen variables) would require a different study design.

Multicollinearity is an important topic in the context of variable selection. Data-driven variable selection methods tend to perform worse if there is a high degree of correlation between the predictors, and their performance will improve the less the predictors are correlated with each other. Before regression analysis is performed in an applied study, the correlations between the independent variables should be checked during initial data analysis [48]. For our simulation study, we have carefully chosen true correlations between the independent variables based on a real correlation matrix from NHANES data. As mentioned in Section 3.2.1, S2 Fig shows the coefficients of determination $R_j^2$ for the regression of each variable $X_j$ on all

other respective variables $X_l$, $l = 1, \ldots, 20$, $l \neq j$. The $R_j^2$ values range from 0 to 0.56, demonstrating differing degrees of dependence between the predictors. These values will be considered when interpreting the simulation results.

In future work, it would be interesting to consider various extensions of our simulation. For example, while we focus on linear and logistic regression in the present protocol, data-driven variable selection is also often used in the context of survival analysis. We plan to conduct a further simulation study comparing different data-driven variable selection methods for Cox regression and the accelerated failure time model.

In the present study, we include several settings where all predictors have true *nonlinear* functional forms, but we nevertheless fit all models with *linear* functional forms; this mimics the frequent misspecification of models in practice. Generally, when fitting a regression model with linear effects, it is advisable to check for misspecification by analyzing the residuals. If misspecification is only mild, then a model with linear effects might still be justifiable. If misspecification is too severe, functional form selection can be performed to account for nonlinear effects, e.g., with spline-based approaches. In future work, our study could be extended by considering the *combination* of variable selection and functional form selection, which is a complex issue [39].

We focus on low-dimensional data in our study. Future studies could compare variable selection methods for high-dimensional data. Finally, our study considers variable selection in a frequentist framework. Future simulation studies could also evaluate Bayesian methods for variable selection.

## Supporting information

**S1 Fig. Correlation network graph.**
(PDF)

**S2 Fig. Absolute standardized regression coefficients plotted against coefficients of determination for each independent variable.**
(PDF)

**S3 Fig. Nonlinear effects.**
(PDF)

**S1 Appendix. Details of the simulation design.**
(PDF)

**S1 Table. Correlation table.**
(PDF)

## Acknowledgments

We would like to thank the members of Topic Group (TG) 2 and the Publications Panel of the STRengthening Analytical Thinking for Observational Studies (STRATOS) initiative for helpful comments. In particular, we thank Willi Sauerbrei, Frank Harrell, Nadja Klein and Harald Binder.

At the time of submission, STRATOS TG2 consisted of the following members (in alphabetical order): Michal Abrahamowicz, Harald Binder, Daniela Dunkler, Frank Harrell, Georg Heinze, Marc Henrion, Michael Kammer, Aris Perperoglou, Willi Sauerbrei, and Matthias Schmid. The group is co-chaired by Georg Heinze (georg.heinze@meduniwien.ac.at), Aris Perperoglou, and Willi Sauerbrei.

## Author Contributions

**Conceptualization:** Theresa Ullmann, Georg Heinze, Daniela Dunkler.

**Funding acquisition:** Georg Heinze, Daniela Dunkler.

**Methodology:** Theresa Ullmann, Georg Heinze, Daniela Dunkler.

**Software:** Theresa Ullmann, Christine Schilhart-Wallisch.

**Supervision:** Georg Heinze, Daniela Dunkler.

**Visualization:** Theresa Ullmann.

**Writing – original draft:** Theresa Ullmann, Georg Heinze, Daniela Dunkler.

**Writing – review & editing:** Theresa Ullmann, Georg Heinze, Lorena Hafermann, Christine Schilhart-Wallisch, Daniela Dunkler.

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
