## [Decision Letter · Decision Letter 0]

16 Apr 2024

PONE-D-24-04044Evaluating variable selection methods for multivariable regression models: a simulation study protocolPLOS ONE

Dear Dr. Dunkler,

Thank you for submitting your manuscript to PLOS ONE. After careful consideration, we feel that it has merit but does not fully meet PLOS ONE’s publication criteria as it currently stands. Therefore, we invite you to submit a revised version of the manuscript that addresses the points raised during the review process. **As pointed out by the reviewer, the setting for the simulations is comparable simple compared to the real world cases, please address this issue seriously.****Also, the authors mentioned when other relevant studies carried out a comparison between their proposed method and the existing methods, the number of the existing methods considered is very limited. However, the authors themselves have not carried out any comprehensive comparison with relevant methods. Please revise the corresponding paragraphs. ****Lastly, when I was a student and my professor told us that the coefficients before g(x) are still called linear coefficients, but the authors named them as nonlinear ones. Please double check it. If I am correct, please revise the corresponding equations and sentences.**

We look forward to receiving your revised manuscript.

Kind regards,

Suyan Tian

Academic Editor

PLOS ONE

Journal Requirements:

2. One of the noted authors is a group or consortium TG2 of the STRATOS initiative. In addition to naming the author group, please list the individual authors and affiliations within this group in the acknowledgments section of your manuscript. Please also indicate clearly a lead author for this group along with a contact email address.

Reviewers' comments:

Reviewer's Responses to Questions

**Comments to the Author**

1. Does the manuscript provide a valid rationale for the proposed study, with clearly identified and justified research questions?

Reviewer #1: No

2. Is the protocol technically sound and planned in a manner that will lead to a meaningful outcome and allow testing the stated hypotheses?

Reviewer #1: No

3. Is the methodology feasible and described in sufficient detail to allow the work to be replicable?

Reviewer #1: No

4. Have the authors described where all data underlying the findings will be made available when the study is complete?

Reviewer #1: Yes

5. Is the manuscript presented in an intelligible fashion and written in standard English?

Reviewer #1: Yes

6. Review Comments to the Author

You may also provide optional suggestions and comments to authors that they might find helpful in planning their study.

**Reviewer #1:** The authors consider variable selection in multivariate regression modeling from simulation perspective. Although the topic is very important in interesting, I am afraid the contribution does not reflect the importance and novelty. I have some specific comments for further consideration:

1- Variable selection first come to notice in (ultra)high-dimensional (HD) cases. In the simulations, I do not see this is highlighted and thus the simulation cannot be a reflective of real-world problems and not practical. In the revision, the authors must exactly specify the relation between the sample size and dimension.

2- In this paper, I did not see any trace of screening. Why did not authors consider selecting important variables based on the marginal correlation of conduct sure screening?

3- The problem of multicollinearity is not considered in variable selection which is absolutely important.

4- The measure R-squared is not a good measure of model fit in HD problems.

5- Form of explanatory variables does not play substantial role; however, the dependent variable plays. I do not see variant responses. Furthermore, in such studies, it is of essential need to consider additive structures in the predictive component. Further investigations for multivariate additive models are needed.

7. PLOS authors have the option to publish the peer review history of their article (what does this mean?). If published, this will include your full peer review and any attached files.

Reviewer #1: No

---

## [Author Response · Author response to Decision Letter 0]

29 May 2024

All comments of the editor and the reviewer are addressed in the PDF file "Response to Reviewers".

---

## [Decision Letter · Decision Letter 1]

26 Jul 2024

Evaluating variable selection methods for multivariable regression models: a simulation study protocol

PONE-D-24-04044R1

Dear Dr. Dunkler,

We’re pleased to inform you that your manuscript has been judged scientifically suitable for publication and will be formally accepted for publication once it meets all outstanding technical requirements.

Kind regards,

Suyan Tian

Academic Editor

PLOS ONE

Additional Editor Comments (optional):

Reviewers' comments:

Reviewer's Responses to Questions

**Comments to the Author**

1. Does the manuscript provide a valid rationale for the proposed study, with clearly identified and justified research questions?

Reviewer #1: Yes

2. Is the protocol technically sound and planned in a manner that will lead to a meaningful outcome and allow testing the stated hypotheses?

Reviewer #1: Yes

3. Is the methodology feasible and described in sufficient detail to allow the work to be replicable?

Reviewer #1: Yes

4. Have the authors described where all data underlying the findings will be made available when the study is complete?

Reviewer #1: No

5. Is the manuscript presented in an intelligible fashion and written in standard English?

Reviewer #1: Yes

6. Review Comments to the Author

You may also provide optional suggestions and comments to authors that they might find helpful in planning their study.

Reviewer #1: I am happy with the revised version. The authors have clearly stated the underlying design of their simulation.

7. PLOS authors have the option to publish the peer review history of their article (what does this mean?). If published, this will include your full peer review and any attached files.

Reviewer #1: No

---

## [Editor Report · Acceptance letter]

31 Jul 2024

PONE-D-24-04044R1 

PLOS ONE

Dear Dr. Dunkler, 

I'm pleased to inform you that your manuscript has been deemed suitable for publication in PLOS ONE. Congratulations! Your manuscript is now being handed over to our production team.

Kind regards, 

on behalf of

Dr. Suyan Tian 

Academic Editor

PLOS ONE